# Metabolic Syndrome and Atrial Fibrillation: Different Entities or Combined Disorders

**DOI:** 10.3390/jpm13091323

**Published:** 2023-08-28

**Authors:** George E. Zakynthinos, Vasiliki Tsolaki, Evangelos Oikonomou, Manolis Vavouranakis, Gerasimos Siasos, Epaminondas Zakynthinos

**Affiliations:** 13rd Department of Cardiology, “Sotiria” Chest Diseases Hospital, Medical School, National and Kapodistrian University of Athens, 11527 Athens, Greece; boikono@gmail.com (E.O.); vavouranakis@gmail.com (M.V.); ger_sias@hotmail.com (G.S.); 2Critical Care Department, University Hospital of Larissa, Faculty of Medicine, University of Thessaly, Mezourlo, 41335 Larissa, Greece; vasotsolaki@yahoo.com (V.T.); ezakynth@yahoo.com (E.Z.); 3Cardiovascular Division, Brigham and Women’s Hospital, Harvard Medical School, Boston, MA 02115, USA

**Keywords:** atrial fibrillation, metabolic syndrome, obesity, insulin resistance, cardiometabolic syndrome

## Abstract

Obesity, hypertension, insulin resistance, and dyslipidemia are all clusters of an entity called “Metabolic Syndrome”. The global trends of this syndrome’s incidence/prevalence continue to increase reciprocally, converting it into a massive epidemic problem in the medical community. Observing the risk factors of atrial fibrillation, a medical condition that is also converted to a scourge, almost all parts of the metabolic syndrome are encountered. In addition, several studies demonstrated a robust correlation between metabolic syndrome and the occurrence of atrial fibrillation. For atrial fibrillation to develop, a combination of the appropriate substrate and a trigger point is necessary. The metabolic syndrome affects the left atrium in a multifactorial way, leading to atrial remodeling, thus providing both the substrate and provoking the trigger needed, which possibly plays a substantial role in the progression of atrial fibrillation. Due to the remodeling, treatment of atrial fibrillation may culminate in pernicious sequelae, such as repeated catheter ablation procedures. A holistic approach of the patient, with simultaneous treatment of both entities, is suggested in order to ensure better outcomes for the patients.

## 1. Introduction

Atrial fibrillation (AF) stands as the most prevalent form of arrhythmia on a global scale, boasting an estimated prevalence of approximately 2–4% among adults—an incidence poised for considerable escalation in the times ahead [1]. Notably, the European Society of Cardiology’s (ESC) task force dedicated to AF diagnosis and management has recently introduced the “Atrial fibrillation Better Care (ABC)” approach. This innovative framework designates the letter ‘C’ to signify Cardiovascular risk factors and concomitant diseases [1]. It is under this guidance that a multidisciplinary and comprehensive approach has been proposed, advocating not only lifestyle modifications and risk factor management but also venturing beyond conventional medical interventions.

The multifaceted dimensions of the ‘C’ encompass central obesity, dyslipidemia, glucose intolerance, and hypertension—constituting both the essence of the ABC approach and collectively forming the framework of metabolic syndrome (MetS) [2]. Notably, registries have garnered evidence supporting a robust correlation between MetS and the prevalence of AF [3,4]. Furthermore, it is discerned that the coexistence of two distinctive clusters within MetS independently contributes to an elevated risk of AF incidence [5].

This review undertakes a twofold objective. Firstly, it endeavors to meticulously assess the interrelationship between AF and MetS. Secondly, it aspires to advance and enrich our comprehension of this specific domain.

## 2. Definitions

### 2.1. Metabolic Syndrome

The first recognizable definition of the MetS was provided by the World Health Organization (WHO) in 1998 as the presence of insulin resistance or fasting plasma glucose > 6.1 mmol/L (110 mg/dL) or 2 h plasma glucose levels > 7.8 mmol (140 mg/dL) (glucose tolerance test) (required) along with two or more of the following: 1. High-Density Lipoprotein (HDL) cholesterol < 0.9 mmol/L (35 mg/dL) in men, <1.0 mmol/L (40 mg/dL) in women 2. Triglycerides > 1.7 mmol/L (150 mg/dL) 3. Waist/hip ratio > 0.9 (men) or >0.85 (women) or Body Mass Index (BMI) > 30 kg/m^2^ 4, Arterial Blood pressure > 140/90 mmHg [6]. Since then, multiple, less recognizable definitions have been proposed [7,8,9], although the more recent definitions of the National Cholesterol Education Program (NCEP) ATP3 2005 [10] and the International Diabetes Federation 2006 (IDF) [11] are the most widely used (Table 1). However, the consensus of several major organizations in 2009 agreed on the newly revised criteria for MetS shown in Table 2 [2]. Since 2009, the definition of MetS has been standardized, and all the above have been used in different studies; innovative trials have proven that fasting glucose is an inadequate parameter due to its inferiority in the younger population and proposed the replacement of elevated fasting glucose with insulin resistance [12,13].

### 2.2. Atrial Fibrillation

According to ESC guidelines, AF is a supraventricular tachyarrhythmia with uncoordinated atrial electrical activation and, consequently, ineffective atrial contraction. The electrocardiographic characteristics of AF are 1. Irregularly irregular R-R intervals (when atrioventricular conduction is not impaired), 2. Absence of distinct repeating P waves, and 3. Irregular atrial activations [1].

## 3. Metabolic Syndrome and Atrial Fibrillation: Is There a Correlation?

The heightened occurrence of AF is concomitant with its correlation to other cardiovascular disorders, such as hypertensive cardiomyopathy, as established in the early 1980s in the renowned Framingham study [14]. The data from this study unveiled a diagnosis of chronic AF in approximately 2% of the population. A prolonged extension of the study over 50 years exposed a substantial surge in AF prevalence. Distinct time periods (1958–1967, 1968–1977, 1978–1987, and 1988–1997) exhibited a fourfold surge in age-adjusted period prevalence from the earliest to the latest interval (1958–1967 versus 1998–2007). This rise underscored the significance of reducing attributable risk factors [15]. Presently, AF prevalence approximates 2–4%, with predictions anticipating a staggering 2.3-fold escalation [1].

Accurate estimation of MetS prevalence remains challenging due to divergent diagnostic criteria among surveys. However, the prevalence of its individual components demonstrates a remarkable increase [16].

Numerous researchers have consistently depicted the robust link between MetS and AF incidence, irrespective of distinct populations and definitions employed across studies (Table 3). Even when the coexistence of major cardiovascular diseases such as coronary heart disease and heart failure is excluded, this correlation remains steadfast (HR, 1.53; 95% CI, 1.35–1.74) [17]. Investigations into the various MetS clusters and their impact on AF incidence yield intriguing findings. Hypertension emerges as the most prevalent component of MetS, while elevated triglycerides represent the least frequent [18,19,20]. However, the contribution of each component to AF incidence varies; some trials indicate the involvement of all MetS components, while others suggest only select clusters are linked to AF occurrence [18,21]. Generally, several trials absolve elevated triglycerides as a factor for AF development [17,22,23]. Conversely, Kim et al. support the notion that only hypertension and obesity may trigger AF [24]. Moreover, the cumulative longitudinal burden of MetS on AF development has been documented. A Korean study categorizing participants based on the number of MetS components demonstrated a positive association with the cumulative number of MetS clusters. This yielded HRs for groups meeting MetS diagnostic criteria with 1, 2, 3, and 4 clusters compared to 0 of 1.18 (95% CI, 1.13–1.24), 1.31 (95% CI, 1.25–1.39), 1.46 (95% CI, 1.38–1.55), and 1.72 (95% CI, 1.63–1.82), *p* < 0.001 [18]. Similar data have emerged from multiple studies [17,25]. Lee et al. sought to substantiate the cumulative effect’s association with AF risk by examining the impact of evolving MetS on the same individual. Their findings revealed that the addition of MetS clusters between examinations incrementally escalated the likelihood of AF development. Participants with 2, 3, and 4 MetS components at reevaluation exhibited increased AF risk by 16%, 32%, and 47%, respectively, compared to those maintaining ≤1 component [HR, 1.164 (95% CI, 1.138–1.192); HR, 1.316 (95% CI, 1.275–1.357); and HR, 1.465 (95% CI, 1.397–1.536), respectively) [25]. Importantly, the reduction of MetS components between examinations corresponded with a decreased AF risk [25], reinforcing the aforementioned premise. Earlier findings prompted Kwon et al. to define a novel entity, termed pro-MetS, encompassing cases where one or two criteria are met. Their analysis showcased elevated AF risk within this group [non-adjusted HRs 2.43 (95% CI 2.313–2.554)] [26]. A meta-analysis of 30,810,460 patients corroborated the strong AF risk-MetS connection, establishing that MetS patients faced elevated AF risk (RR 1.57; 95% CI 1.40–1.77; Figure 1 in comparison to those without MetS. Notably, all MetS components, except elevated triglycerides, were associated with heightened AF incidence in line with this meta-analysis, albeit with substantial heterogeneity (I2 = 97%) [3].

## 4. A Correlation Exists: But Why?

To comprehend the link between MetS and AF and to establish a cohesive understanding of the mechanisms underlying AF development, it is imperative to first elucidate the pathogenetic factors leading to AF occurrence. An array of parameters contributes to the pivotal atrial transformations necessary for the initiation of AF, encompassing elements like inflammation, ion-channel modifications, hypocontractility, fatty infiltration, vascular remodeling, and fibrosis [1]. The crux of these mechanisms can be encapsulated by two pivotal terms: “trigger” and “substrate”.

The “trigger” takes on the role of an instigator, initiating atrial fibrillation through rapid and repetitive ectopic beats. These triggers are predominantly located at the junctional interface between the pulmonary veins and the left atrium, although they can manifest throughout both the left and right atria. These triggers can manifest as rapidly firing ectopic beats, various types of arrhythmias, and factors inducing atrial stretch, ischemia, or autonomic disorientation—examples include mitral regurgitation and myocardial infarction [30].

On the other hand, the “fibrotic substrate” constitutes the principal pathophysiological component in the maintenance of atrial fibrillation. It catalyzes structural or electrical remodeling of the atria. Disturbances in myocyte action potential or conduction between atrial cells contribute to electrical remodeling, while atrial dilation and fibrosis are key players in instigating structural changes [30]. Consequently, MetS must either function as a trigger point or furnish the essential substrate to induce AF. Remarkably, all components of MetS have demonstrated correlations with AF, albeit through distinct mechanisms (Figure 1).

### 4.1. Obesity

Atrial fibrillation is markedly more prevalent among obese individuals compared to those with a BMI below 30.0 kg/m^2^ [31,32]. Remarkably, even within the latter group, an elevated level of abdominal fat has been associated with an increased risk of AF occurrence [32]. Notably, an expanded body size during early adulthood, elevated BMI in midlife, and weight gain from the age of 20 to midlife have all been linked to an augmented likelihood of future AF development. This highlights that increased BMI at any stage of life can potentially serve as a risk factor for AF [33]. Individuals classified as obese face nearly a 50% heightened risk of developing AF when compared to their non-obese counterparts [34].

Obesity exerts a twofold influence by altering both cardiac hemodynamics and heart morphology. This leads to an amplified blood volume, increased cardiac output, remodeling of cardiac chambers, and an augmentation of epicardial fat [35]. Cardiac remodeling precipitated by obesity encompasses the emergence of left ventricular hypertrophy, resulting in diastolic dysfunction and pronounced enlargement of the left atrium [35,36]. These effects, in conjunction with the altered hemodynamics—such as heightened stroke volume and pulmonary pressures—establish an environment conducive to the initiation of AF [36]. The potential for echocardiographic left atrial volumetric enlargement over a decade is almost 2.4 times higher in obese individuals [37]. Moreover, pericardial adipose tissue significantly contributes to the burgeoning frequency of AF [38]. Of particular note, epicardial fat secretes adipocytokines, including Activin-1, which incites fibroblast proliferation and amplifies fibrosis within myocardial tissue [39]. This phenomenon plays a role in shaping the substrate required for AF.

Mahajan et al. highlighted the impact of obesity on cardiac tissue in sheep, demonstrating electroanatomical changes. Significantly reduced mean conduction velocity was observed in the obese group (LA 1.18 ± 0.04 m/s vs. 1.58 ± 0.04 m/s, *p* < 0.001) compared to the normal group. While the mean total voltage remained unchanged, distinct regional voltage patterns emerged in the left atrium of the obese and control groups, primarily due to a substantial reduction in posterior LA voltage (3.7 ± 2.3 mV vs. 5.5 ± 2.3 mV; *p* < 0.001) [40]. These findings were recently corroborated in men by the same research group [conduction velocity: 0.86 ± 0.31 m/s vs. 1.26 ± 0.29 m/s; *p* < 0.001]. In the obese group, 13.9% of all points in the left atrium exhibited low voltage, compared to 3.4% in the reference group (*p* < 0.001) [41]. These electrical changes contribute to the creation and maintenance of the substrate required for AF occurrence.

### 4.2. Hypertension

Hypertension contributes to around 14% of all cases of AF, and among AF patients, over 70% have hypertension. Hypertension independently amplifies the risk of AF progression and leads to the emergence of adverse effects associated with AF [42]. The impact of long-standing hypertension on the heart is well-documented, resulting in elevated cardiac filling pressures and diastolic dysfunction. A study from Norway established a link between diastolic dysfunction and AF [43]. Notably, this connection is applicable in cases with both left atrial enlargement and without it, where elevated left ventricular filling pressures primarily drive AF [44]. However, the most significant structural alteration attributed to hypertension is the enlargement of the left atrium (LA), influenced by various mechanisms encompassing hemodynamic and electrical remodeling, neurohormonal activation, and inflammation [45]. Electrical remodeling is also evident in individuals with hypertension, characterized by extensive areas of double potentials, fractionated signals, and varying and slower regional activation times within the atria, leading to atrial conduction delays [46]. Another noteworthy aspect of hypertension is the role of the renin-angiotensin-aldosterone system (RAAS), which becomes highly activated in hypertensive individuals. Experiments conducted on mice revealed that elevated levels of angiotensin II trigger both electrophysiological and structural modifications in the heart, along with increased fibrosis in the left atrium [47,48]. This fibrosis, as mentioned earlier, constitutes a significant factor in the development of AF [49]. Moreover, an alternate pathway through which hypertension may lead to fibrosis and subsequently to AF involves the extensive activation of the inflammatory process [50].

### 4.3. Insulin Resistance/Diabetes

Individuals with either type of diabetes have a twofold higher prevalence of AF, a figure that steadily rises as the severity of the disease and its microvascular complications progress. The presence of silent AF episodes is highly probable due to autonomic dysfunction [1]. Similar to hypertension, diabetes also involves structural and electrical remodeling, autonomic dysregulation, and inflammation. The renin–angiotensin–aldosterone system (RAAS) is enhanced, and elevated levels of angiotensin II are linked to diabetes, contributing to atrial fibrosis [51]. Another contributing factor in diabetes is the increased production of advanced glycation end products (AGEs) and their receptors, potentially exacerbating atrial scarring and fibrosis and thus contributing to the substrate for AF [52]. Consequently, patients develop diabetic cardiomyopathy characterized by diastolic dysfunction, which creates conditions within the atria conducive to AF occurrence [53]. Diabetic patients exhibit delayed conduction velocities and atrial emptying [54], intra-atrial electromechanical delay [55], prolonged action potentials, and abnormal atrial voltage [56]. These observations might be explained by the pathological expression of gap-junction proteins known as connexins [57]. Autonomic abnormalities, including sympathetic upregulation leading to sympathetic denervation and an imbalance in the autonomic nervous system—termed diabetic neuropathy—also impact the heart and can contribute to AF [58,59]. Inflammation and oxidative stress further contribute to the proarrhythmic state of diabetes. Disturbed mitochondrial function prevents the elimination of reactive oxygen species (ROS), leading to the activation of inflammation pathways evidenced by elevated inflammatory markers [60]. Additionally, hypoglycemia, with its accompanying sympathetic activation and fluctuations in blood glucose levels, can reinforce myocardial fibrosis and elevate oxidative stress, forming unique key mediators of the arrhythmic substrate in diabetes [61,62].

### 4.4. Dyslipidemia

The role of dyslipidemia in the occurrence of AF has not yet been fully elucidated. Low HDL cholesterol has been associated with an increased risk of AF, although elevated triglycerides have not shown a similar correlation [3]. However, findings from a Japanese cohort indicated that the relationship between low HDL and AF risk exists only in women, not men [63]. In contrast to the preceding clusters of MetS, the underlying mechanisms connecting dyslipidemia to AF have not been extensively investigated. Elevated blood lipids create an inflammatory environment and increase oxidative stress [64], potentially contributing to AF development. A comparison between patients with and without AF revealed that the AF group had 1.6-fold higher plasma triglycerides and increased inflammation markers [65]. Increased myocardial triglyceride content (MTGC), as measured by magnetic resonance spectroscopy, has been linked to diastolic dysfunction, although the precise connection between MTGC and plasma triglycerides remains unclear [66]. Recent studies have suggested a connection between postprandial, very low-density lipoprotein (VLDL), composed of triglycerides, and atrial remodeling in MetS patients [67]. Further support for this observation comes from new data demonstrating that a surplus of triglycerides within VLDL leads to atrial enlargement and disturbances in PR duration. In MetS patients, the left atrium diameter and volume were larger compared to non-MetS individuals (LA diameter: non-MetS 3.2 ± 0.3 cm vs. MetS-off statin 4.4 ± 0.4 cm vs. MetS-on statin 4.3 ± 0.3 cm, *p* < 0.0001; LA maximum volume: non-MetS 45.2 ± 9.5 mL vs. MetS-off statin 81.9 ± 13.9 mL vs. MetS-on statin 77.5 ± 18.9 mL, *p* < 0.0001; LA minimum volume: non-MetS 28.1 ± 7.5 mL vs. MetS-off statin 39.3 ± 10.8 mL vs. MetS-on statin 42.0 ± 11.3 mL, *p* = 0.0065). Additionally, atrioventricular conduction, as measured by PR interval, was prolonged in MetS patients (176.1 ± 19.0 ms vs. 156.2 ± 15.4 ms, *p* = 0.0014) [68]. Therefore, it is plausible that dyslipidemia contributes to structural and electrical changes in the cardiac chambers, yet further research is warranted in this area.

### 4.5. Metabolic Syndrome

It is readily apparent that when these risk factors converge within the framework of “Metabolic syndrome,” they profoundly disrupt the proper functioning of the left atrium, precipitating the occurrence of AF through a multitude of mechanisms. Overall, there is insufficient evidence to substantiate the direct association of MetS as a distinct entity with AF. Instead, much of the research has focused on the individual components of MetS in relation to AF. A rat study revealed that obesity compounds the atrial arrhythmogenic phenotype in hypertensive rats, exacerbating interstitial atrial fibrotic changes, conduction velocities, and left atrial emptying [69]. This lends credence to the idea that MetS, as an integrated entity, can indeed instigate AF. Additionally, evidence suggests that MetS patients experience both structural alterations—such as left ventricular and left atrial remodeling—and electrical changes (PR interval in MetS group vs. non-MetS: 167.6 ± 20.0 msec vs. 156.2 ± 24.9 msec, *p* = 0.0064) [67,68]. Furthermore, both MetS and AF are characterized by inflammation, and their coexistence has been associated with even greater levels of inflammation, potentially introducing another mechanism that leads to fibrosis and AF development [70]. Beyond inflammatory markers, other biomarkers dysregulated in MetS—such as adiponectin, leptin, ghrelin, uric acid, and OxLDL—appear to contribute to the initiation and progression of AF. Adiponectin appears to mitigate cardiac chamber remodeling [71]. Leptin-mediated pathways impact Angiotensin metabolism, thus contributing to AF [72]. Moreover, since leptin regulates calcium homeostasis, it significantly influences electrophysiological pathways [73]. Lastly, MetS disrupts autonomic tone, with the degree of impairment corresponding to the number of MetS clusters [74]. This observation suggests that MetS could potentially play a role not only in the formation of substrate but also in triggering AF (Figure 2).

The human heart, alongside skeletal muscles, kidneys, and the brain, is a highly energy-consuming organ, necessitating significant amounts of adenosine triphosphate (ATP) molecules for proper function. Approximately 60% of the required energy is derived from fatty acid (FA) metabolism through β-oxidation [75]. However, this process is accompanied by a reduction in the NADH/NAD+ system within mitochondria, leading to the generation of electrons that enter the electron transport chain (ETC) to produce ATP. In situations where β-oxidation becomes overactive, or ATP levels decrease, an excess of electrons is generated, resulting in the conversion of these electrons into superoxide radicals (ROS) [76]. Metabolic syndrome has been demonstrated to lead to a condition in which the heart predominantly relies on FA for ATP production, imposing excessive stress on mitochondria and causing damage through various mechanisms [77]. The malfunctioning ATP production mechanisms lead to a decreased ATP-to-O_2_ consumption ratio, triggering hyperactivity of the cardiac muscle, subsequent hypertrophy, and diastolic dysfunction. Moreover, mitochondria play a crucial role in maintaining Ca^2+^ homeostasis. Thus, mitochondrial dysfunction results in disruptions and oscillations of intracellular Ca^2+^ levels, potentially inducing arrhythmias [78]. Lastly, due to the incapacity of mitochondria to generate ATP and their shift towards ROS production in the ETC, substantial quantities of ROS are released into circulation [77]. As mentioned earlier, ROS triggers inflammation, leading to structural remodeling of the heart as previously described. Consequently, recent findings underscore the significance of mitophagy—the process that selectively eliminates damaged mitochondria from cells, facilitating their degradation by lysosomes and maintaining mitochondrial homeostasis—in the pathophysiology of Metabolic Syndrome (MetS). There is a suggestion for its potential therapeutic application in the future [79].

## 5. Are These Patients Suitable for Catheter Ablation?

The management of atrial fibrillation (AF) has undergone significant transformation in the past two decades, with an increasing reliance on catheter ablation as a favored approach. The successful adoption of ablation as a viable treatment hinges on its capacity to either catalyze or substantially diminish AF recurrence while ensuring safety and enhancing quality of life. Achieving these objectives hinges not only on the efficacy of the procedure itself but also on the reduction of AF risk factors [80].

Initial data suggested that obesity, a noteworthy risk factor, did not significantly impact the outcomes of catheter ablation [81]. However, recent investigations have demonstrated that only individuals with a BMI greater than 35 kg/m^2^ experience effects on ablation success and complications. The role of BMI remains complex and contradictory, with lower BMI associated with paroxysmal AF and higher BMI correlated with persistent AF. This implies that elevated BMI could be a detrimental prognostic factor for AF ablation, primarily due to the augmented likelihood of ablation failure in persistent AF cases compared to paroxysmal AF [82]. Notably, weight loss might contribute to improved quality of life in these patients rather than a reduction in AF burden [83]. Conversely, a comprehensive meta-analysis revealed that every 5-unit increase in BMI was associated with nearly a 30% elevated risk of post-ablation AF occurrence [84]. The influence of hypertension on AF ablation outcomes is still unclear. In contrast to earlier findings, a German registry demonstrated that hypertension did not significantly impact the long-term results of AF ablation [85]. However, conditions such as diabetes and impaired fasting glucose not only contribute to AF recurrence post-ablation but also alter the atrial remodeling process following the procedure [56]. The question remains: How does Metabolic Syndrome (MetS) as a unified entity affect AF ablation outcomes? Firstly, catheter ablation can be safely conducted in patients with MetS, with no substantial difference in the incidence of complications between MetS and non-MetS patients (1.86% vs. 2.42%, *p* = 0.621) [86]. It is worth noting that this observation did not reach statistical significance in any of the existing studies, likely due to the limited number of complications, thus warranting further research. In terms of AF recurrence, MetS has been associated with a considerable increase in post-ablation AF [87,88]. Initially, MetS was linked with the recurrence of non-paroxysmal AF (150 [46%] vs. 257 [35%], *p* < 0.002), whereas its impact on paroxysmal AF was less significant (39 [25%] vs. 62 [22%] in group 2, *p* = 0.295) within MetS vs. non-MetS groups, respectively. Nonetheless, the quality of life for both types of AF improved during a 1-year follow-up [87]. However, another study by Xia et al. demonstrated a robust correlation between MetS and paroxysmal AF (freedom from AF: MetS vs. non-MetS: 50.0% vs. 74.2%, log-rank *p* < 0.01). MetS was associated with an increased risk of AF recurrence (HR = 2.15, 95% CI: 1.207–3.841, *p* = 0.009) [88]. A meta-analysis further highlighted that younger individuals with MetS exhibited an even higher risk of recurrence (RR, 3.03; 95% CI, 1.70–5.40) [89]. Additionally, MetS emerged as the sole predictor for AF reappearance two years post-ablation [90], a theory supported by an analysis of risk factors contributing to unsuccessful ablation outcomes. MetS, obesity, and hypertension were identified as the only modifiable factors [91]. When combined with obstructive sleep apnea, MetS significantly elevated the probability of AF recurrence post-ablation [92]. Consequently, all AF risk factors, including MetS, must be meticulously considered by medical practitioners during treatment planning. In conclusion, while catheter ablation is a recommended option for patients with MetS, enduring results necessitate a holistic patient-centered approach alongside vigorous interventions to address all modifiable AF risk factors.

## 6. What Drugs to Use?

### 6.1. Antiarrhythmic

Regrettably, there is a paucity of evidence and recommendations regarding the effectiveness of various antiarrhythmic drugs in patients with Metabolic Syndrome (MetS). Amiodarone, classified as a class III antiarrhythmic drug, has garnered criticism for its potential acute hepatotoxicity when rapidly infused intravenously and chronic hepatotoxicity when administered orally [93]. Nevertheless, in most instances, these adverse effects are largely reversible, and only in rare cases do they escalate to cirrhosis [93]. It is important to note that MetS has also been associated with hepatic damage, particularly non-alcoholic fatty liver disease [94]. Consequently, the use of amiodarone in individuals with MetS may augment the risk of drug-induced hepatotoxicity. Nonetheless, current guidelines recommend judicious use of amiodarone when deemed appropriate, accompanied by vigilant monitoring of aminotransferase levels [93]. Similarly, dronedarone, a newer class III antiarrhythmic drug, has been linked to potential liver damage. However, unlike amiodarone, the precise extent of its impact remains uncharted [93]. Encouraging data from a sub-analysis of the ATHENA and EURIDIS/ADONIS studies shed light on the use of dronedarone in patients with diabetes [95]. In the context of obese patients with AF, a study by Ornelas-Loredo et al. explored the efficacy of sodium channel blocker antiarrhythmic drugs (Class I) and their response in relation to obesity. Their findings indicated that obesity, both in patients and mice, was associated with an increased likelihood of poor response to such drugs, a trend more pronounced for class I drugs (class I vs. class III AAD: OR, 4.54; 95% Wald CI, 1.84–11.2; *p* = 0.001). The authors suggested that reduced expression of sodium channels in obesity might underlie this phenomenon and proposed considering class III drugs for obesity-related AF treatment [96]. Given these insights, it is plausible that class III antiarrhythmic drugs could be considered a favorable option for AF management in MetS patients, with a caveat for potential hepatic harm. Notably, amiodarone tends to accumulate in adipose tissue, and in obese patients with an increased adipose mass, the drug’s pharmacokinetics are influenced, leading to reduced drug clearance [96]. In light of this observation, personalized amiodarone dosing may be warranted, particularly in obese individuals. This avenue holds promise for future investigations to explore in greater depth. Finally, a discrepancy occurs regarding b-blockers. This class of drugs has been framed for weight gain and glucose intolerance [97,98], as well as for provocation of new-onset diabetes [99]. However, the Glycemic Effects in Diabetes Mellitus Carvedilol—Metoprolol Comparison in Hypertensive study (GEMINI) demonstrated that b-blockers a-adrenergic effects, such as carvedilol, improve insulin sensitivity, slower progression of microalbuminuria in diabetes and have a safer metabolic profile than cardioselective b-blockers, such as metoprolol [100,101,102]. The improvement of metabolic parameters was also shown by the YESTONO study for nebivolol due to induced nitric oxide synthesis leading to vasodilating effects [103]. As a result, the revised ESC guidelines for hypertension of 2018 suggest the usage of these types of b-blockers in patients with MetS [104].

### 6.2. Anticoagulants

Over the last few decades, direct oral anticoagulants (DOACs) have been increasingly replacing older drugs like warfarin in the management of atrial fibrillation. Their overall effectiveness and safety have not been thoroughly examined in the context of Metabolic Syndrome (MetS) but rather in isolation for its individual clusters. The effectiveness and safety of rivaroxaban in diabetic patients have shown similar or even superior results compared to warfarin, resulting in fewer bleeding events [105,106]. Notably, DOACs demonstrate fewer instances of hypoglycemia when used alongside antidiabetic treatments, presenting a more favorable safety profile compared to warfarin [107]. A meta-analysis comparing DOACs for both effectiveness and safety in diabetic patients indicated that lower-dose Dabigatran (110 mg BID) might be the better option, followed by rivaroxaban at its full dose (20 mg once per day) [108]. Similar to diabetes, DOACs have proven to be effective and generally safe for use in obese individuals [109]. In a study comparing rivaroxaban and apixaban against warfarin, rivaroxaban demonstrated a safer profile than apixaban, both effectively protecting patients from AF-related complications [110]. Moreover, separate research indicated that apixaban outperforms warfarin in terms of both safety and efficacy [111]. Recent studies even confirmed the safety and efficacy of apixaban and rivaroxaban in extremely overweight patients (BMI > 50 kg/m2) [112]. However, due to an elevated bleeding risk associated with edoxaban, it is recommended to opt for DOACs with a more favorable balance between efficacy and adverse effects [113]. Interestingly, in contrast to diabetic patients, dabigatran in obese individuals has shown an increased risk of gastrointestinal bleeding and might not be recommended when other DOACs are available [114]. Apixaban has demonstrated superiority over other DOACs in obese patients with both AF and heart failure [115]. Given that MetS is a complex condition with varying mechanisms affecting the body in distinct ways, determining the optimal treatment for patients with co-existing diabetes and obesity remains a challenge. Based on existing data, the utilization of rivaroxaban or apixaban appears to be suitable, as neither drug has shown inferiority compared to warfarin or other DOACs. Nevertheless, this conclusion is preliminary, and more randomized clinical trials specifically focusing on MetS and anticoagulants are warranted to provide clarity on these pertinent questions. (Table 4).

### 6.3. Holistic Approach

Focusing solely on intervening at the source of AF through techniques like catheter ablation or antiarrhythmic drugs addresses the “trigger” point of AF while leaving the proarrhythmic substrate untouched. To provide effective relief from AF symptoms in patients with Metabolic Syndrome (MetS), our approach must target both the trigger and the substrate. Consequently, our strategy should also address the various clusters of MetS. Weight reduction through intensive interventions not only alleviates AF burden and severity but also leads to the remodeling of heart chambers, resulting in a reduction of left atrial (LA) volume [116]. This reversal of effects extends beyond the heart’s dimensions and even alters the type of AF and its natural progression. Remarkably, weight loss exceeding 10% has been strongly associated with transitioning from persistent to paroxysmal AF (odds ratio 4.3, 95% confidence interval 2.7–6.8; *p* < 0.001) [117]. The LEGACY Study categorized patients into three groups based on weight loss and noted that AF frequency, severity, and duration significantly improved in the first two groups compared to the third (*p* < 0.001). Additionally, more patients in the first two groups remained free from AF (45.5% in Group 1, 22.2% in Group 2, and 13.4% in Group 3; *p* < 0.001). Notably, weight loss also positively impacted other clusters of MetS, improving glycemic control, blood pressure management, and lipid profiles [118]. Surgical weight reduction, such as bariatric surgery, has shown superior effects on reducing AF incidence compared to medical treatments [119]. Even after catheter ablation, maintaining a weight reduction of up to 10% was associated with a 27% reduced likelihood of AF recurrence [120]. Hypertension and AF frequently coexist, placing patients at a higher risk for complications. Optimal blood pressure levels for these patients have been identified as systolic blood pressure of 120–129 mmHg and diastolic blood pressure < 80 mmHg, as levels outside these ranges are associated with increased adverse cardiovascular effects [121]. Among antihypertensive drugs, telmisartan has shown encouraging results in reducing AF recurrence compared to amlodipine and ramipril (amlodipine: 44.2%, ramipril: 25.5%, telmisartan: 12.9%, vs. amlodipine, *p* < 0.01 and vs. ramipril, *p* < 0.05), despite similar blood pressure reduction with the other drugs [122]. Therefore, selecting the appropriate antihypertensive drug is crucial, although complete data on the best antihypertensive for AF patients is still lacking. Controlling lipid levels is also beneficial for preventing AF recurrence. Statins for lipid control have demonstrated a reduction in AF occurrence [123,124], although intensive lipid-lowering may not affect triglycerides and small lipoproteins. The impact of lowering triglycerides in MetS patients with AF remains uncertain. Glycemic control is pivotal in AF management. A 10% reduction in HbA1c levels 12 months before catheter ablation reduces the likelihood of AF recurrence by 30% [125]. Ideally, maintaining HbA1c levels below 6.9% increases the chances of successful ablation [126]. Papazoglou et al. emphasized the role of glycemic control on AF recurrence and highlighted specific drugs that affect it. Biguanides, thiazolidinediones, secretagogues, and sodium-glucose cotransporter 2 (SGLT-2) inhibitors reduce the risk of AF recurrence, while glucagon-like peptide-1 (GLP-1) receptor agonists have a neutral effect on AF. Conversely, insulin and dipeptidyl peptidase 4 inhibitors have an inverse relationship with AF [127]. The ARREST-AF Cohort Study has shown that aggressive reduction of multiple risk factors significantly enhances ablation success rates and considerably increases arrhythmia-free survival (87% in the Risk Factor Management group compared to 17.8% in the control group, *p* < 0.001) [80]. Risk factor management also leads to a 38% reduction in the need for initial ablation and a 20% decrease in the need for redo ablation [128].

## 7. Key Points

Metabolic syndrome significantly elevates the incidence of atrial fibrillation.Cardiac chamber structural and electrical remodeling, autonomic imbalance, inflammation, oxidative stress, and fibrosis constitute the primary pathways through which metabolic syndrome contributes to atrial fibrillation.For patients with both metabolic syndrome and atrial fibrillation, a comprehensive approach is advised, encompassing the management of all syndrome clusters and subsequent strategies for rhythm regulation.The imperative for clinical trials is evident, aiming to elucidate the optimal antiarrhythmic drugs and anticoagulants for this patient population.

## 8. Conclusions

Metabolic syndrome is a multifaceted collection of conditions that have been on the rise in prevalence in recent times. Atrial fibrillation, likewise, is a condition for which our knowledge and understanding continue to evolve, with an expanding array of techniques for prevention and treatment. Devising an effective approach for addressing the simultaneous occurrence of these two entities remains a complex challenge for healthcare providers, and achieving optimal management necessitates the collaboration of various medical specialties. The growing requirement for meticulously designed randomized clinical trials is paramount in addressing the knowledge gaps that have emerged in the management of patients with both Metabolic Syndrome and atrial fibrillation.

## Figures and Tables

**Figure 1 jpm-13-01323-f001:**
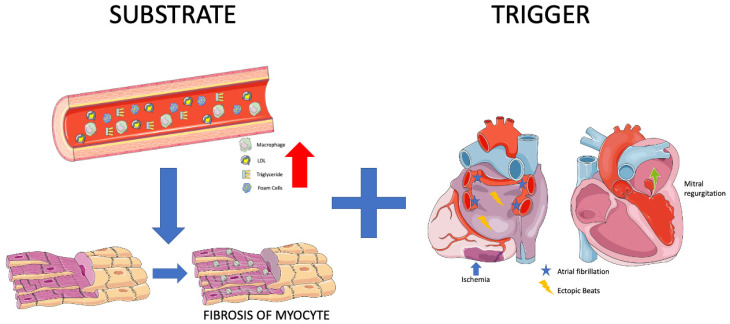
Substrate and Trigger. (Parts of the figure were drawn by using pictures from Servier Medical Art. Servier Medical Art by Servier is licensed under a Creative Commons Attribution 3.0 Unported License (https://creativecommons.org/licenses/by/3.0/, accessed on 20 August 2023).

**Figure 2 jpm-13-01323-f002:**
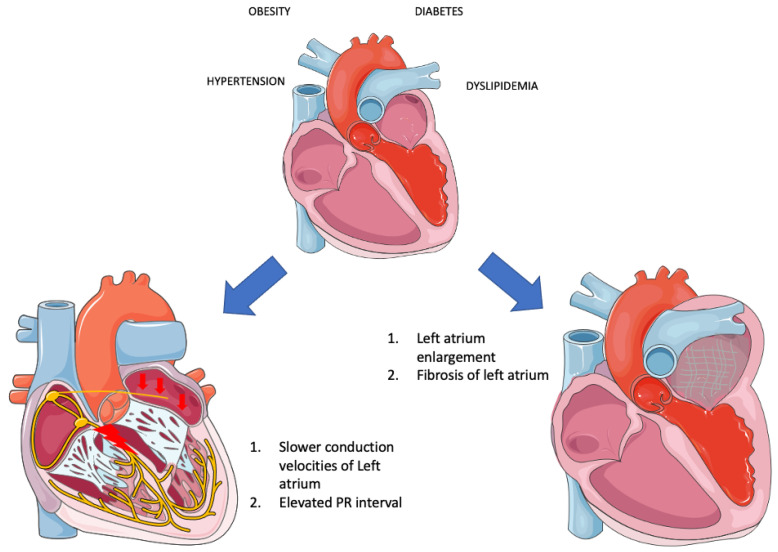
Electrical and structural remodeling in metabolic syndrome (Parts of the figure were drawn by using pictures from Servier Medical Art. Servier Medical Art by Servier is licensed under a Creative Commons Attribution 3.0 Unported License (https://creativecommons.org/licenses/by/3.0/, accessed on 20 August 2023).

**Table 1 jpm-13-01323-t001:** Revised Criteria for Clinical Diagnosis of the Metabolic Syndrome.

Revised Criteria for Clinical Diagnosis of the Metabolic Syndrome	
Measure	Categorical Cut Points
Elevated waist circumference *	Population- and country-specific definitions
Elevated triglycerides (drug treatment for elevated triglycerides is an alternate indicator †)	150 mg/dL (1.7 mmol/L)
Reduced HDL-C (drug treatment for reduced HDL-C is an alternate indicator †)	<40 mg/dL (1.0 mmol/L) in males; <50 mg/dL (1.3 mmol/L) in women
Elevated blood pressure (antihypertensive drug treatment in a patient with a history of hypertension is an alternate indicator)	Systolic 130 and/or diastolic 85 mm Hg
Elevated fasting glucose ‡ (drug treatment of elevated glucose is an alternate indicator)	100 mg/dL

HDL-C indicates high-density lipoprotein cholesterol. * It is recommended that the IDF cut points be used for non-Europeans and either the IDF or AHA/NHLBI cut points used for people of European origin until more data are available. † The most used drugs for elevated triglycerides and reduced HDL-C are fibrates and nicotinic acid. A patient taking 1 of these drugs can be presumed to have high triglycerides and low HDL-C. High-dose −3 fatty acids presume high triglycerides. ‡ Most patients with type 2 diabetes mellitus will have the metabolic syndrome by the proposed criteria.

**Table 2 jpm-13-01323-t002:** Diagnostic Criteria of Metabolic Syndrome.

Definitions of Metabolic Syndrome
National Cholesterol Education Program (NCEP) ATP3 2005
Blood glucose greater than 5.6 mmol/L (100 mg/dL) or drug treatment for elevated blood glucose HDL cholesterol < 1.0 mmol/L (40 mg/dL) in men, <1.3 mmol/L (50 mg/dL) in women or drug treatment for low HDL-C Blood triglycerides > 1.7 mmol/L (150 mg/dL) or drug treatment for elevated triglycerides Waist > 102 cm (men) or >88 cm (women) Blood pressure > 130/85 mmHg or drug treatment for hypertension
**International Diabetes Federation 2006 (IDF)**
Central obesity (defined as waist circumference ≥ 94 cm for Europid men and ≥80 cm for Europid women, with ethnicity-specific values for other groups) plus any two of the following four factors: Raised TG level: >150 mg/dL (1.7 mmol/L) or specific treatment for this lipid abnormality. Reduced HDL cholesterol: <40 mg/dL (1.0 mmol/L) in males and <50 mg/dL (1.3 mmol/L) in females, or specific treatment for this lipid abnormality Raised blood pressure: systolic BP ≥ 130 or diastolic BP ≥ 85 mm Hg, or treatment of previously diagnosed hypertension. Raised fasting plasma glucose (FPG) ≥ 100 mg/dL (5.6 mmol/L) or previously diagnosed type 2 diabetes. If above 5.6 mmol/L or 100 mg/dL, OGTT is strongly recommended but is not necessary to define the presence of the syndrome.

BP, Blood Pressure; FPG, Fasting Plasma Glucose; HDL, High-Density Lipoprotein; OGTT, Oral Glucose Tolerance Test; TG, Triglycerides.

**Table 3 jpm-13-01323-t003:** Characteristics and results of the studies included in the review.

Studies	Design	Total Cases	Population	MetS Definition	AF-MetS (% of Total Population)	Findings
Hyo-Jeong Ahn et al. [18], 2021	Retrospective Cohort Study	2.985.189	Korean (2009–2013) (NHID) (aged > 20)	NCEP-ATPIII + modified waist circumference (WC) criteria of the Korean Society for the Study of Obesity	−14.1	positive association of AF with the cumulative number of MetS criteria 1.46 (1.38–1.55), and 1.72 (1.63–1.82), *p* for trend < 0.001. HR for 3 and 4 criteria, respectively
Pastori et al. [27], 2021	Prospective Cohort Study	1.735	Italian (mean age 75.1) Patients with a history of AF	NCEP-ATPIII	100–49	MetS and NAFLD were more frequently affected by persistent/permanent AF AF combined with MetS showed a higher risk for CVEs
Lee et al. [25], 2021	Retrospective Cohort Study	7.565.531	Korean (2008–2009) (NHID) (aged >20)	AHA/NHLBI	1.79–36.9	AF risk was higher by 31% in the MM group [hazard ratio (HR), 1.308; 95% CI, 1.290–1.327], 26% in the MH group (HR, 1.259; 95% CI, 1.238–1.280), and 16% in the HM group (HR, 1.155; 95% CI, 1.134–1.178) compared with the HH group, respectively
Wang et al. [28], 2020	Prospective Cohort Study	81.092	Chinese (2006–2007) (Kailuan study) (aged 18–98)	NCEP-ATPIII	0.3–29.8	MS and a high hs-CRP level were associated with higher AF risk (HR = 1.61; 95% CI 1.08–2.41; *p* = 0.019)
Choe et al. [21], 2019	Retrospective Cohort Study	22.896.663	Korean (2009–2012) (NHID) (aged > 40)	NCEP-ATPIII	0.98–27.4	HR for incident AF in patients with MetS was 1.38 (95% confidence interval [CI] 1.36–1.39) compared to those without MetS
Kwon et al. [26], 2019	Retrospective Cohort Study	7.830.602	Korean (2009–2016) (NHID) (aged 30–69)	NCEP-ATPIII	0.26–15.9	Incidence of AF 0.12% in the normal group and 0.53% in the MetS group
Kim et al. [24], 2018	Retrospective Cohort Study	21.981	Korean (2003–2008) (University Hospital of Ulsan) (mean age 46)	IDF	0.8–11.5	MetS was associated with an increased risk of AF. Age-adjusted HR for AF in subjects with MetS was 1.62 (95% CI 1.08–2.44, *p* = 0.02)
Nyström et al. [19], 2015	Prospective Cohort Study	4.021	Swedish (1997–1999)	Revised MetS criteria of IDF [11]	7.1–27.6	37.9% of the AF group had MetS vs. 26.8% of the non-AF group had MetS
Vyssoulis et al. [5], 2013	Prospective Cohort Study	15.075	Greek (1988–2010) (aged > 40) Patients with hypertension	NCEP-ATPIII AHA/NHLBI WHO IDF GISSI Score	Not mentioned—from 31.7 to 47.8, according to the definition used	Presence of MS in patients with hypertension was constantly associated with a higher incidence of AF in all groups (*p* < 0.001). Odds ratio 1.61 to 1.99, depending on the definition of MS used
Chamberlain et al. [17], 2010	Prospective Cohort Study	15.094	Americans (1987–1989) Atherosclerosis Risk in Communities (ARIC) Study Two groups: Black race and white race (aged 45–64)	AHA/NHLBI	8.2–41.1	HR for AF among individuals with, compared to those without, the MetSyn was 1.67 (95% CI, 1.49–1.87) in both races
Tang et al. [20], 2009	Retrospective Cohort Study	741	Chinese (2005–2007) (Mean age 55.8)	NCEP-ATPIII	100–46.3	Higher prevalence of MetS in AF than that in Chinese adults (46.3% vs. 16.5%, *p* < 0.001)
Watanabe et al. [22], 2008	Prospective Cohort Study	28.449	Japanese (1996–1998) (aged > 20)	NCEP-ATPIII AHA/NHLBI	0.9–13 (NCEP-ATPIII) 16 (AHA/NHLBI)	HR for developing AF in patients with METS was 1.88 (95% CI, 1.4–2.52) for NCEP-ATPIII and 1.61 (95% CI, 1.21–2.15) for AHA/NHLBI
Umetanani et al. [23], 2007	Prospective Cohort Study	592	Japanese (2001–2005) (mean age 63)	NCEP-ATPIII	5–21	MetS was a risk factor for PAF/PAFL independently from other parameters OR 2.8, 95% confidence interval (CI) 1.3–6.2, *p* < 0.01)
Echadidi et al. [29], 2007	Retrospective Cohort Study	5.085	Canadians (2000–2004) (mean age 64) After CABG	NCEP-ATPIII	27–46	Incidence of AF in patients with MetS was 29% and 26% in patients without MetS (*p* = 0.01)

AHA/NHLBI, American Heart Association (AHA) and the National Heart, Lung, and Blood Institute; AF, Atrial Fibrillation; ARIC, Atherosclerosis Risk in Communities; CABG, Coronary Artery Bypass Graft; CI, Confidence Interval; hs-CRP, high sensitive C reactive Protein; MetS, Metabolic Syndrome; NCEP-ATPIII, NAFLD, Non-Alcoholic Fatty Liver Disease; National Cholesterol Education Program Adult Treatment Panel III, PAF/PAFL, Paroxysmal Atrial Fibrillation/Paroxysmal Atrial Flatter; IDF GISSI; HR, Hazard Ratio.

**Table 4 jpm-13-01323-t004:** Drugs for Atrial Fibrillation in patients with Metabolic Syndrome.

Antiarrhythmic Drugs	
Amiodarone (Class III—Potassium channel blockers) [93,94]	Preferred in MetS Monitoring of aminotransferases Dosage modification in obesity
Dronedarone (Class III—Potassium channel blockers) [95]	Less hepatotoxic Less studied drug Effective in diabetics
Class I—Sodium channel blockers [96]	Reduce expression of sodium channels in obesity Reduce efficacy in obesity
Class II—B blockers [97,98,99,100,101,102,103,104]	Cardioelective b-blockers: weight gain, glucose intolerance, induce new-onset diabetes -> not suggested B-blockers with a-adrenergic effect (carvedilol): improve metabolic parameters -> suggested Vasodilating b-blockers (Nebivolol): improve metabolic parameters-> suggested
Class IV—Nondihydropyridine Calcium channel blockers	Data exist only for Dihydropyridine calcium channel blockers that lack antiarrhythmic effect
**Anticoagulants**	
Dabigatran [105,106,107,108]	Low dose (110 mg BID) best for diabetics Higher risk of bleeding in obesity
Rivaroxaban [105,106,107,108,109,110]	Second best option for diabetics/full dose (20 mg daily) Safer profile than Apixaban Safe for extremely overweight
Apixaban [110,111,112,114,115]	Safe in obesity Safe for extremely overweight Best option for obese patients with heart failure and AF
Edoxaban [113]	Last option, due to high bleeding risk

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
