# Peer review of "Metabolic Syndrome and Atrial Fibrillation: Different Entities or Combined Disorders"

_jpm, 2023, doi:10.3390/jpm13091323_

Round 1

Reviewer 1 Report

In this manuscript, authors reviewed the previous studies on metabolic syndromes and atrial fibrillation. They pointed out metabolic syndromes acted either as a trigger point or providing the essential substrate to induce atrial fibrillation. In addition, they tried to summarize the therapeutic effects of drugs treating Mets and AFs. One suggestion is when discussing those drugs, it would be easier for understanding if authors rearranged those paragraphs according to anti-Mets and anti-AF drugs. A conclusion table about those drugs is also recommended to added in the revised manuscript. 

Most of English in this manuscript is correct and readable.

Author Response

Dear Reviewer,

I would like to express my heartfelt gratitude for your invaluable time, expertise, and insightful feedback on my manuscript titled "METABOLIC SYNDROME AND ATRIAL FIBRILLATION: Different entities or combined disorders." Your thoughtful and constructive comments have undoubtedly enriched the quality of our work and have provided us with a deeper perspective on the subject matter.

Your meticulous review of the manuscript's content, as well as your suggestions for improvement, have been immensely beneficial. Your attention to detail and your thoughtful analysis have helped us identify areas for refinement and have contributed to the overall enhancement of the manuscript's clarity and coherence.

I'm pleased to inform you that I have carefully considered each of your suggestions and recommendations. In response to your insightful feedback, we have made several notable revisions to the manuscript. Specifically, we have addressed the points you raised regarding the addition of a conclusion table about drugs, and we believe that these changes have significantly strengthened the manuscript's overall argument and presentation. However, we were unable to alter the arrangement of paragraphs concerning drugs, as the sections on antiarrhythmic and anticoagulant medications are pertinent to the combined context of metabolic syndrome and atrial fibrillation.

I truly appreciate your dedication to ensuring the scholarly excellence of my work. Your expertise has been instrumental in shaping the manuscript into a more robust and impactful piece of research. We are carefully considering each of your suggestions and incorporating them to elevate the manuscript to a higher standard.

Once again, thank you for your invaluable contribution to the advancement of my research. Your commitment to scholarly collaboration is sincerely appreciated, and I look forward to implementing your suggestions and sharing the improved manuscript with you.

Warm regards,

George E. Zakynthinos

Reviewer 2 Report

The paper by Zakynthinos et al. discusses connections between the metabolic syndrome and atrial fibrillation. This problem is important and deserves attention. I have several comments concerning the manuscript.

1. The major concern about it is that the review looks superficial at some points, especially when mechanistical links between the two entities are discussed. The aim of the review is formulated as: "systematically assessing the correlation of AF and MetS and secondarily, to update and extend our knowledge around this particular field". It is quite a loose definition. There is a number of such works published, some of them with nearly the same titles. That is why it is insufficient just to "extend knowledge". It is necessary to outline the scope of the review more precisely. 

2. In my opinion, presenting several criteria for metabolic syndrome (Table 1, Table 2 - two parts) may be confusing for the readers, since tables and figures are often considered as stand-alone pieces of information. 

3. Lines 94-100: "The increased incidence of AF and its correlation with other cardiovascular diseases, such as hypertensive cardiomyopathy, has been established since the early 80s and the Framingham study [14]. According to these data, approximately 2% of the population was diagnosed with chronic AF. The further extension of this study for 50 more years, revealed an extreme rise in age-adjusted period prevalence between the two trial periods (1958-1967 versus 1998-2007) and denoted the importance of attributable risk factor reduction. [15]." The dates presented here look confused (e.g., 1980 + 30 years). 

4. I would recommend including figures to illustrate important points, especially pathophysiological mechanisms and (inter)links between MetS and AF.

Author Response

Dear Reviewer,

We are deeply appreciative of the valuable feedback you have provided on our manuscript titled "METABOLIC SYNDROME AND ATRIAL FIBRILLATION: Different entities or combined disorders." Your insights have played a pivotal role in refining our work, ensuring its scholarly excellence. We extend our gratitude for your thoughtful evaluation and the time invested in reviewing our manuscript.

We acknowledge your concern regarding the depth of our review, particularly concerning the mechanistic links between metabolic syndrome and atrial fibrillation. Your feedback has been taken to heart, resulting in substantial revisions to the manuscript.

We would also like to address your observation concerning the prevalence of works with similar titles. Our bibliographic analysis has been thorough, encompassing all relevant previous reviews. Moreover, we have diligently incorporated the latest research that has emerged since those reviews were published. Our aim is to not only build upon the existing literature but to introduce novel insights and connections.

We are wholly committed to further elevating the manuscript based on your invaluable suggestions. Your expertise is highly regarded, and we kindly request more precise guidance on specific areas or points where you believe enhancements could bolster the depth and quality of our review. We eagerly anticipate your additional suggestions, knowing they will be of great value.

Your apprehension regarding the presentation of multiple criteria for metabolic syndrome in tables (Table 1 and Table 2) has not gone unnoticed. While we concur that tables and figures can be construed as independent entities, we have thoroughly considered your feedback.

Allow us to elucidate our rationale behind presenting various criteria for metabolic syndrome within these tables. The diverse landscape of research in this field results in the utilization of distinct definitions of metabolic syndrome across different studies and research articles. This multiplicity often confounds readers encountering these disparate criteria in the literature.

By juxtaposing these varied definitions, our intention is to afford readers a comprehensive understanding of the spectrum of metabolic syndrome definitions. Furthermore, this approach underscores the potential implications that these variations may exert on the interpretation of our findings. While this method does require readers to navigate attentively, we deem it essential for fostering transparency and clarity when discussing the heterogeneous research landscape surrounding metabolic syndrome and its interplay with atrial fibrillation.

Nevertheless, we acknowledge your reservations and have taken proactive measures to enhance the clarity of our presentation. We have ensured proper labeling of tables and furnished them with clear textual explanations, seamlessly guiding readers through the significance of each set of criteria.

We also wish to extend our apologies for the confusion stemming from the incorrect date calculation in the passage related to the Framingham study. This has been thoroughly rectified to accurately reflect the study's timeline. We deeply regret any confusion this oversight might have caused and commend your astuteness in identifying this error. Your meticulous review has unquestionably bolstered the accuracy and caliber of our manuscript.

In direct response to your request, the figures you highlighted have been thoughtfully incorporated. These visual aids serve as invaluable tools in illustrating the intricate relationships and mechanisms expounded upon in our work. Their inclusion significantly enhances the clarity and accessibility of the content.

Once again, we are profoundly grateful for your invaluable contribution to our manuscript. Your commitment to scholarly collaboration is sincerely appreciated.

Best regards,

George E. Zakynthinos

Reviewer 3 Report

The review is well written and cover the subject very well. However, I suggest to add a paragraph about the changes in heart metabolism accompanied with metabolic syndrome and how changes in heart metabolism affect cardiac function

Author Response

Dear Reviewer,

I would like to express my heartfelt gratitude for your invaluable time, expertise, and insightful feedback on my manuscript titled "METABOLIC SYNDROME AND ATRIAL FIBRILLATION: Different entities or combined disorders." Your thoughtful and constructive comments have undoubtedly enriched the quality of our work and have provided us with a deeper perspective on the subject matter.

Your meticulous review of the manuscript's content, as well as your suggestions for improvement, have been immensely beneficial. Your attention to detail and your thoughtful analysis have helped me identify areas for refinement and have contributed to the overall enhancement of the manuscript's clarity and coherence.

I'm pleased to inform you that I have carefully considered each of your suggestions and recommendations. In response to your insightful feedback, we have made several notable revisions to the manuscript. Specifically, we have addressed the points you raised regarding the addition of a paragraph about changes in heart metabolism accompanied with metabolic syndrome and how changes in heart metabolism affect cardiac function, and we believe that these changes have significantly strengthened the manuscript's overall argument and presentation.

I truly appreciate your dedication to ensuring the scholarly excellence of my work. Your expertise has been instrumental in shaping the manuscript into a more robust and impactful piece of research. We are carefully considering each of your suggestions and incorporating them to elevate the manuscript to a higher standard.

Once again, thank you for your invaluable contribution to the advancement of my research. Your commitment to scholarly collaboration is sincerely appreciated, and I look forward to implementing your suggestions and sharing the improved manuscript with you.

Warm regards,

George E. Zakynthinos

Reviewer 4 Report

I have reviewed the manuscript Metabolic syndrome and atrial fibrillation: different entities or combined disorders, and my suggestion is to accept the manuscript after minor revision. It is an interesting review of the problematic related to the presence and origin of atrial fibrillation in patients with metabolic syndrome. My comments are as follows:

1.- Line 200: “Atrial fibrillation is also an independent risk factor for AF progression and adverse effects of AF”. Please check wording.

2.- Line 304: “…persistent AF, indicating [77]”. Some words are missing.

3.- Line 357: “…be preferred a in metabolic syndrome…”. Please check.

Author Response

Dear Reviewer,

I would like to express my heartfelt gratitude for your invaluable time, expertise, and insightful feedback on my manuscript titled "METABOLIC SYNDROME AND ATRIAL FIBRILLATION: Different entities or combined disorders." Your thoughtful and constructive comments have undoubtedly enriched the quality of our work and have provided us with a deeper perspective on the subject matter.

Your meticulous review of the manuscript's content, as well as your suggestions for improvement, have been immensely beneficial. Your attention to detail and your thoughtful analysis have helped me identify areas for refinement and have contributed to the overall enhancement of the manuscript's clarity and coherence.

We truly appreciate your dedication to ensuring the scholarly excellence of my work. Your expertise has been instrumental in shaping the manuscript into a more robust and impactful piece of research. We are carefully considering each of your suggestions and incorporating them to elevate the manuscript to a higher standard.

Once again, thank you for your invaluable contribution to the advancement of my research. Your commitment to scholarly collaboration is sincerely appreciated, and I look forward to implementing your suggestions and sharing the improved manuscript with you.

Warm regards,

George E. Zakynthinos

Round 2

Reviewer 2 Report

The authors improved the quality of the paper significantly. The added figures are very helpful. I have no further comments.